# A Simple Preparation Route for Bio-Phenol MQ Silicone Resin via the Hydrosilylation Method and its Autonomic Antibacterial Property

**DOI:** 10.3390/polym11091389

**Published:** 2019-08-23

**Authors:** Jianye Ji, Xin Ge, Weijie Liang, Ruiyuan Liang, Xiaoyan Pang, Ruoling Liu, Shuyi Wen, Jiaqi Sun, Xunjun Chen, Jianfang Ge

**Affiliations:** 1Guangdong Engineering Research Center of Silicone Electronic Fine Chemicals, College of Chemistry and Chemical Engineering, Zhongkai University of Agriculture and Engineering, Guangzhou 510225, China; 2School of Materials and Energy, Guangdong University of Technology, Guangzhou 510006, China; 3School of Materials Science and Engineering, Northwestern Polytechnical University, Xi’an 710072, China; 4GanSu Yinguang Juyin Chemical Co., Ltd., Baiyin 730900, China

**Keywords:** Bio-Phenol MQ silicone resin, hydrosilylation, modified, thermal stability, antibacterial property

## Abstract

MQ silicone resins represent a broad range of hydrolytic condensation products of monofunctional silane (M units) and tetrafunctional silane (Q units). In this work, a Bio-Phenol MQ silicone resin (BPMQ) was designed and synthesized by the hydrosilylation of hydrogen containing MQ silicone resin and eugenol in the presence of chloroplatinic acid. The structure, thermal property, and antibacterial property against *Escherichia coli* of the modified MQ silicone resin were investigated. The results showed that BPMQ has been prepared successfully, and the thermal stability of this modified polymer improved significantly because of the introduction of phenyl in eugenol. The temperature at the maximum degradation rate increased from 250 °C to 422.5 °C, and the residual yields mass left at 600 °C were increased from 2.0% to 28.3%. In addition, its antibacterial property against *Escherichia coli* was also enhanced markedly without adding any other antimicrobial agents. This improved performance is ascribed to special functional groups in the structure of eugenol. The BPMQ polymer is expected to be applied to pressure-sensitive adhesives and silicone rubber products for the biomedical field due to its reinforcing effect and antioxidant quality.

## 1. Introduction

MQ silicone resin is an organic–inorganic hybrid material with a three-dimensional structure consisting of tetra-functional siloxane (SiO_4/2_, Q units) and mono-functional siloxane (R_3_SiO_1/2_, M units). The core of MQ silicone resin is a cage-type SiO_2_ (Q units) with high density, while the shell is M units with low density, and organic/inorganic groups such as methyl, aryl, vinyl, or silicon hydrogen bonds exist in M units [1,2,3]. This organic–inorganic distinctive structure gives the MQ silicone resin excellent thermal stability [4], high weather resistance [5], high chemical resistance [6], etc. MQ silicone resin can be used as a constituent of a high-temperature coating, as a tackifier for a silicone pressure-sensitive adhesive, and as a filler for liquid silicone rubber [7]. MQ silicon resin is used as a reinforcing agent in a polymer matrix, which can significantly improve the mechanical properties and thermal stability of the composites, and further improve their application value [8].

However, it is not enough to improve the mechanical properties and thermal stability of the polymers when using MQ silicone resin as a reinforcing filler. For instance, beyond improving the adhesive properties and thermal stability of adhesive, we still need to ensure the bacteriostatic and biocidal properties when MQ silicone resin is applied to silicone pressure-sensitive adhesives for medical applications. Simultaneously, if the silicone rubber can be given certain bactericidal and bactericidal performance by choosing MQ silicone resin as a reinforcement agent, it will further improve the application value and broaden the application scope of the silicone rubber. With the rapid development of medical and hygiene products, the demand for silicone pressure-sensitive adhesives and silicone rubber products is increasing [9]. For instance, the silicone pressure-sensitive adhesive has been widely used in medical tape, elastic bandages, wound paste, film of surgical incision, female sanitary products, diapers, etc. [10]. In the same way, silicone rubber has been widely used in medical bandages, artificial organs, and wearable devices [11]. There are more stringent health and safety requirements for these products because they come into direct contact with human skin. Therefore, as one of the essential constituents of silicone pressure-sensitive adhesives and an excellent reinforcement filler of liquid silicone rubber, MQ silicone resin must have certain bacteriostatic properties.

Eugenol (4-allyl-2-methoxyphenol) is a bio-phenol that derives from plants; it is one of the main constituents of plant essential oils, such as camphor oil, clove oil, cinnamon leaf oil, and nutmeg oil [12]. Its molecular structure is characterized by phenol hydroxyl, a carbon methoxy group, and an allyl group. Due to this distinctive structure, eugenol has some special biological and chemical characteristics, such as antibacterial activity, anticancer activity, antioxidant activity, thermal stability, chemical reactivity, and biological compatibility [13,14]. Manrique [15] compared the survival behavior of *Staphylococcus carnosus*, *Listeria innocua*, *Escherichia coli*, and *Pseudomonas fluorescens* treated with sequential doses or one-time treatments of eugenol, and the results showed that sequential half-doses were bacteriostatic regardless of the application times and a higher antimicrobial efficacy on the *Escherichia coli* was observed. Pathirana et al. [16] investigated the antibacterial activity of clove essential oil and eugenol on seven Gram-negative and nine Gram-positive fish pathogenic bacteria isolated from cultured olive flounder in Korea, which indicated that clove essential oil and eugenol inhibited the growth of both Gram-negative and Gram-positive bacteria, and a positive correlation was observed between multiple antibiotic resistance index values and the minimum inhibitory concentration values of clove essential oil and eugenol. Dai et al. [17] prepared the eugenol-based organic coatings through the introduction of eugenol and the results showed that the coatings exhibited good thermal stability, high hardness, outstanding adhesion, and excellent solvent resistance. Furthermore, it is reported that eugenol has anticancer and antioxidant properties [18,19]. Moreover, research shows that the phenolic hydroxyl group plays a pivotal role in antimicrobial activity, and methoxy on the ortho carbon could also enhance the antibacterial activity [12,20].

In the hope of not destroying the phenolic hydroxyl and ortho-methoxy of eugenol, we attempted to combine the antibacterial nature of eugenol and the excellent physical and chemical properties of MQ silicone resin. This will further broaden the application of MQ silicone resin to medical-grade silicone pressure-sensitive adhesive and food-grade silicone rubber. For this reason, bio-Phenol MQ Silicone Resin (BPMQ) was synthesized via a hydrosilylation reaction of hydrogen containing MQ silicone resin (HMQ) and eugenol. Afterwards, the structure, thermal properties, and antibacterial activity of the polymer were investigated. It is expected that we can obtain a novel MQ silicone resin with excellent thermal stability and certain antibacterial activity.

## 2. Materials and Methods

### 2.1. Materials

1,1,3,3-tetramethyldisiloxane (TMDSO, AR), hexamethyldisiloxane (HMD, AR) and tetraethyl orthosilicate (TEOS, AR) were provided by Shanghai Hansi Chemical Industry Co., Ltd. (Shanghai, China). Ethanol (C_2_H_5_OH, AR), hydrochloric acid (HCl, AR) and toluene (98 wt %, AR) were purchased from Tianjin Baishi Chemical Industry Co., Ltd. (Tianjin, China). Eugenol (98 wt %, AR) was obtained from Guangdong Tongcai New Material Corporation (Guangzhou, China). Chloroplatinic acid (H_2_PtCl_6_, 5000 mg/Kg, AR) was supplied by Tianjin Maisike Chemical Industry Co., Ltd. (Tianjin, China). Bromo-acetic acid (98.5 wt %, AR), carbon tetrachloride (CCl_4_, 95 wt %, AR), iodine bromide (IBr, 98 wt %, AR), potassium iodide (KI, 100 g/L, AR), sodium thiosulfate (Na_2_S_2_O_3_, 0.1 mol/L, AR) and starch solution (1.0 wt %, AR) were purchased from Shanghai Aladdin Bio-Chem Technology Co., Ltd. (Shanghai, China).

### 2.2. Synthesis of the HMQ

The HMQ is prepared via the hydrolysis and condensation reaction of TEOS, MM, and TMDSO, according to the preparation procedure reported by Xu et al. with a minor amendment [21]. Deionized water (40.0 mL), toluene (50.0 mL), HCl (6.0 mL), ethanol (10.0 mL), TMDSO (30.25 g), and MM (4.10 g) were added into a 500-mL three-necked flask, and the temperature held constant at 25 °C. We allowed the mixtures to react for about 30 min, and then TEOS (104.2 g) was added to the flask drop by drop. The temperature was raised to 60 °C and stirring continued for 3 h. After that, we used toluene (150 mL) to extract the products and deionized water to wash the products until they reached a neutral pH. Finally, we distilled the polymer–toluene solution at 110 °C for about 2 h to obtain the HMQ product. The mass fraction of hydrogen in HMQ was measured, and the results showed that content of hydrogen is 0.5% (Table 1).

### 2.3. Synthesis of Bio-Phenol MQ Silicone Resin

BPMQ was prepared through a hydrosilylation reaction of HMQ and eugenol. HMQ (50.0 g), eugenol (39.1 g), and toluene (6.0 mL) were added to a 250 mL three-necked flask equipped with a nitrogen inlet, magnetic stirrer, and condenser. After venting the nitrogen for about 15 min, the hydrosilylation catalyst H_2_PtCI_6_ (0.3 g) was added to the flask. Then, we stirred the mixture for 5 h at 75 °C. Finally, we distilled the toluene and low boiling components at 110 °C for about 2 h to obtain the BPMQ polymer.

### 2.4. Characterizations and Measurements

Content of hydrogen groups in HMQ: the mass fraction of hydrogen groups in HMQ was measured via iodometric titration according to a procedure in the literature [22]. Firstly, the sample was dissolved into a 250 mL iodine flask containing a CCl_4_ solution (20.0 mL) and bromo-acetic acid solution (10.0 mL). Secondly, IBr (10.0 mL) was added to the iodine flask and shaken evenly. After that, the reaction continued for about 1 h in the dark. Thirdly, KI solution (15 mL) was added into the flask and shaken for 5 min, then distilled water (40 mL) was used to wash the iodine flask. Fourth, we titrated the solution until it was light yellow with Na_2_S_2_O_3_ of 0.1 mol/L, then a starch solution (1.0 mL) was added and we continued to titrate the solution until the blue coloration disappeared. In the same way, the blank experiment was carried out. The mass fraction of hydrogen groups in HMQ was determined by the following equation:(1)$$\Phi =\frac{1.008\times 0.001\times 0.5\times C\times \left(V_{0}-V_{1}\right)}{M}\times 100\%,$$
where *Φ* is the mass fraction of hydrogen groups in HMQ (%); *C* is the concentration of Na_2_S_2_O_3_ (mol/L); *V*_0_ is the volume of Na_2_S_2_O_3_ consumed by the blank experiment; *V*_1_ is the volume of Na_2_S_2_O_3_ consumed by the experimental group; the constant 1.008 is the molar mass of hydrogen group; the constant 0.001 is the unit conversion constant between mL with L; the constant 0.5 is the coefficient of mole balance; and *M* is the mass of the HMQ sample.

FT-IR: The structure of the HMQ and BPMQ samples was characterized by VERTEX70 Fourier transform infrared (FT-IR) spectra (Shimadzu Corporation, Kyoto, Japan). Each sample was scanned from 450 to 4000 cm^−1^ and averaged over eight times.

^1^H-NMR: The structures of the HMQ and BPMQ were characterized at 25 °C with an Av500 hydrogen nuclear magnetic resonance spectrometer (Bruker Corporation, Karlsruhe, Germany) at a frequency of 500 MHz. We used deuterated chloroform (CDCl_3_) as the solvent and tetramethylsilane as an internal reference. 

^29^Si NMR: The ^29^Si NMR analysis of the HMQ and BPMQ samples was conducted with a Bruker AVANCE AV 400 MHz spectrometer at 25 °C, with CDCl_3_ as the solvent. 

Gel permeation chromatography (GPC): Molecular weights and molecular weight distribution of HMQ and BPMQ were determined by GPC using an Agilent 1260 infinity LC system (Agilent Technologies, Palo Alto, CA, USA), equipped with two PLgel MixedC and one PLgel Mixed-D columns and a refractive index detector. The operation was performed at 25 ± 1 °C, using HPLC-grade THF as the eluent at the flow rate of 1 mL/min and polystyrene as the molecular weight reference. 

Viscosity: The viscosity of HMQ and BPMQ was measured on a Haake Mars II rotational rheometer (Haake, Karlsruhe, BW, Germany) under a steady shear flow at room temperature. The apparent viscosity was obtained at a given shear rate of 10 S^−1^.

Thermogravimetric analysis (TGA): Thermogravimetric analysis was carried out on a Mettler Toledo TG thermal analyzer (Mettler Toledo, Zurich, Switzerland) at a flow rate of 20 mL/min in nitrogen, the sample was heated from 40 °C to 800 °C at a rate of 10 °C/min.

Antimicrobial activity analysis: The antimicrobial activity was measured according to a spread plate method, using *Escherichia coli* as the bacterial strain and agar as the nutrients. Agar plates containing *Escherichia coli* were put into the biochemical incubator at 25 °C for 24 h. The amounts of *Escherichia coli* used for the experiment were optimized according to a procedure in the literature [12]. To more intuitively assess the antimicrobial effect of HMQ, PBMQ and eugenol, the same mass fractions of HMQ and BPMQ (equivalent to 20% of the weight of agar) were added to the agar plates. Likewise, we kept the amount of eugenol identical to that in BPMQ. Optical images of the cultivation plate were taken with a 12-megapixel iPhone 6s mobile phone (Apple Inc., Cupertino, CA, USA), and the number of *Escherichia coli* colonies was determined by manual counting.

## 3. Results and Discussion

### 3.1. Structural Analysis

HMQ was synthesized first via hydrolysis and a condensation reaction. To determine the additional amount of HMQ needed for a further reaction, the content of hydrogen groups in HMQ was measured by chemical titration. The results showed that the mass fraction of hydrogen in HMQ was 0.5%. Then, BPMQ was prepared via the hydrosilylation reaction of HMQ and eugenol, using H_2_PtCl_6_ as an additional catalyst. Theoretically, this reaction of Si–H and vinyl is in the molar ratio of 1:1, but to overcome side effects caused by the lively reaction of Si–H in HMQ, a slightly excess molar ratio of HMQ to eugenol was used to promote the reaction more completely, and a molar ratio of 1.02:1 of the value of Si–H/–CH=CH_2_ was chosen in this reaction [23]. The preparation routes of HMQ and BPMQ are shown in Figure 1.

The structures of HMQ and BPMQ were characterized by FT-IR. As shown in Figure 2, in the curve of HMQ, the sharp peaks at 2960 and 2903 cm^−1^ are assigned to the stretching vibration of C–H group on Si–CH_3_. The strong absorption peak at 2140 cm^−1^ is attributed to Si–H stretching vibration in the molecular structure of HMQ [24]. In addition, the sharp peaks at 1418, 1260, and 841 cm^−1^ are ascribed to the characteristic absorption peaks of Si–CH_3_, while the peak located at 1080 cm^−1^ is a characteristic symmetric stretching absorption peak of Si–O–Si. These indicate that HMQ has been synthesized successfully [25]. In the curve of PBMQ, the broad peak at 3550 cm^−1^ corresponds to the stretching vibration of phenol hydroxyl. The characteristic absorption peaks at 3060 and 3010 cm^−1^ are ascribed to the stretching vibration of the C–H group on phenyl [26], while the peaks at 1610 and 1520 cm^−1^ are C–C stretching vibration of benzene. The peak at 1370 cm^−1^ corresponds to C–H in C–CH_3_, which was introduced from eugenol. Moreover, the characteristic absorption peak of Si–H at 2140 cm^−1^ has disappeared, and the symmetric stretching vibrations of C–H on Si–CH_2_ formed during the hydrosilylation at 2850 cm^−1^ were found. These indicate that BPMQ has been prepared successfully through the hydrosilylation reaction of HMQ and eugenol.

The molecular formulas of HMQ and BPMQ and ^1^H NMR spectra of HMQ and BPMQ are shown in Figure 2. In the ^1^H NMR spectra curve of HMQ, only two chemical shifts of protons can be observed. The characteristic peaks at about 0–0.15 ppm corresponded to the protons on –CH_3_ of terminal M units, and the signal at 4.71 ppm was attributed to the proton peaks of Si–H. Compared with HMQ, except for the characteristic signals of protons on –CH_3_, it multiple different signals of proton peaks are seen in the 1H NMR spectra of BPMQ. The chemical shifts at 6.71, 6.53, and 6.17 ppm belong to the chemical shifts of protons in phenyl region [27]. The signals at about 5.39 and 3.72 ppm are the characteristic absorption peaks for phenol hydroxyl and ortho-methoxy of eugenol, respectively [28]. In addition, the chemical shift at 0.42–0.63 ppm was assigned to the proton peaks of C–CH_3_, and the characteristic peak at 1.70–1.79 ppm corresponded to the proton of tertiary carbon (C–H). However, strong signals for the protons of the Si–CH_2_CH_2_CH_2_–Si groups were also observed at chemical shifts of 1.01–1.13, 1.41–1.55, and 2.19–2.25 ppm, which indicated that the hydrosilylation of HMQ and eugenol was carried out as a result of α-addition and β-addition products, dominated by β-addition [8,29]. Meanwhile, the chemical shifts of Si–H occurring at 4~5 ppm and vinyl of eugenol at 5~6 ppm were not found. This proved that the reaction was complete and BPMQ was prepared successfully.

The structure of HMQ and BPMQ were also determined by ^13^C NMR spectra, as shown in Figure 3c; in the curve of HMQ, only a single chemical shift was found at about −1~0.8 ppm, assigned to the carbon atoms of silicon methyl. In the curve of BPMQ, the expected chemical signals for carbon atoms in the phenyl moiety corresponded to about 109~144 ppm. The characteristic peak for ortho-methoxy of eugenol was at 54 ppm, the carbon atoms’ characteristic absorption peaks of methyl in tertiary carbon (CH_3_) were at about 13 ppm, and characteristic signals of the tertiary carbon (CH) at 16 ppm. In addition, chemical shifts for Si–CH_2_CH_2_CH_2_–Si at 19~20 and 23~24 ppm have also been found. This further illustrated that BPMQ has been synthesized successfully through the hydrosilylation reaction of the Markovnikov and Anti-Markovnikov rule [30].

^29^Si NMR spectra were a useful tool to reflect the skeleton structure of the organic silicone polymer. Figure 4 presents the ^29^Si NMR spectra of HMQ and BPMQ. In the curve of HMQ, the signal at about –110 ppm corresponds to the Q (SiO_4/2_) units. Chemical shifts at about −5.0 and 11.8 ppm were assigned to M(R_3_SiO_1/2_) and ^H^M(R_1_R_2_SiO_1/2_) units, respectively [31]. Compared with the ^29^SiNMR spectra curve of HMQ, the characteristic peak for ^H^M(R_1_R_2_SiO_1/2_) units at about −5 ppm could not be found, and peaks at about 15 ppm for Si–CH_2_CH_2_ CH_2_–Si of M units appeared, which indicated that Si–H in HMQ had been consumed completely and the BPMQ was prepared successfully.

The molecular weights of HMQ and BPMQ are listed in Table 1. The weight average molecular weights of HMQ and BPMQ were 2000 and 2600 dal.mol^−1^, respectively. The average molecular weights of HMQ and BPMQ were 1400 and 1800 dal.mol^−1^, respectively, with a similar polydispersity index of 1.43 and 1.44. Obviously, the molecular weights of BPMQ were slightly higher than those of HMQ, which can be attributed to the introduction of the eugenol micromolecule. Moreover, this slight change in molecular weights indicates that side reactions such as self-condensation of HMQ or dehydrogenation of HMQ and BPMQ barely existed in the system. 

### 3.2. Viscosity Analysis

The effect of the introduction of eugenol was investigated by a viscosity analysis of HMQ and BPMQ. As depicted in Table 1, the dynamic viscosity of BPMQ was 455 mpa.s at room temperature, which was a little higher than that at 440 mpa.s for HMQ. The viscosity of polymers is related to their molecular weight, so a higher molecular weight generally results in higher viscosity under the same condition [32]. This slight viscosity increase for the polymer reflects a limited increase in their molecular weight and almost no side effects in this reaction.

### 3.3. Thermal Property

The weight loss of HMQ and BPMQ was measured to evaluate the effect of eugenol on enhancing the thermal stability of BPMQ. Figure 5 presents the TG and DTG curves of the samples. Two degradation stages were observed in the overall thermal degradation of HMQ in nitrogen. The first stage is likely due to the degradation of organic groups (M units) of HMQ at the range of 100–410 °C. The temperature at the maximum degradation rate of HMQ in this step was 250 °C. The second degradation stage of HMQ was in the temperature range of 410–600 °C, which could be attributed to the decomposition of the Si–O network structure in Q units of HMQ [4,32]. HMQ has almost finished degradation when the temperature reaches 600 °C, and possesses residual yields of 2.0%. Three degradation steps were found in the overall thermal decomposition of BPMQ. The first step is mainly due to the degradation of some smaller molecules, such as toluene reagent, small amounts of volatile water, and residual eugenol at the range of 50–220 °C. The second decomposition step of BPMQ can be observed at the temperature range of 220–490 °C, and is also mainly ascribed to the degradation of M units in BPMQ. Meanwhile, the temperature at the maximum degradation rate of BPMQ in this stage was 422.5 °C, which was far higher than that (250 °C) for HMQ. This is likely due to the introduction of a benzene structure in the eugenol [33]. Moreover, BPMQ has almost finished degradation at about 600 °C, similar to the temperature at the maximum degradation of HMQ. This demonstrates that HMQ and BPMQ have the similar structure. Simultaneously, the residual yields of BPMQ at 600 °C was 28.3%, which was about 26.3% higher than that of HMQ. It was concluded that the introduction of eugenol can enhance the thermal stability of HMQ.

### 3.4. Antibacterial Property

The antimicrobial activity is displayed in Figure 6, showing the blank group (a), HMQ (b), eugenol (c) and BPMQ (d). In the experiment, the antimicrobial activity was assessed by choosing *Escherichia coli* as a template and agar as the nutrient solution. In order to more intuitively assess the antimicrobial effect of HMQ, PBMQ, and eugenol, the same mass fraction of HMQ and BPMQ was added to the agar plates; simultaneously, we kept the amount of eugenol identical to that in BPMQ. It can be observed that the agar culture dish of the blank experiment was almost completely covered in *Escherichia coli* colonies after incubation for one day (Figure 6a). Similarly, the agar culture dish with the presence of 20 wt % HMQ was also completely covered in *Escherichia coli* colonies after 24 h (Figure 6b), which illustrated that HMQ has almost no antimicrobial activity. In addition, there were almost no visible *Escherichia coli* colonies on the culture dish containing 20 wt % eugenol (Figure 6c), which indicates that eugenol exhibits superior antibiotic activity. Compared with Figure 6a,d, the number of *Escherichia coli* colonies in the agar culture dish containing the same amount of BPMQ was far lower than in the control group and the one with HMQ. This shows that the introduction of eugenol enhanced the antimicrobial activity of HMQ due to the unique structure (containing phenolic hydroxyl and an ortho-methoxy group) of eugenol [34,35]. Moreover, according to some reports, the antibacterial activity of bio-phenol siloxane may be related to the electrical, hydrophobic, and spatial network structure [12,36]. The result shows that BPMQ had favorable autonomic antimicrobial activity.

## 4. Conclusions

Bio-Phenol MQ silicone resin was prepared through a hydrosilylation reaction of hydrogen containing MQ silicone resin and eugenol. Compared to the unmodified MQ silicone resin, the thermal stability of this modified polymer improved significantly because of the introduction of phenyl in eugenol. The temperature at the maximum degradation rate rose from 250 °C to 422.5 °C, and the residual yields at 600 °C were increased from 2.0% to 28.3%. In addition, its antibacterial property against *Escherichia coli* was also enhanced markedly without adding other antimicrobial agents. The bio-phenol MQ silicone resin had autonomic antimicrobial activity due to special functional groups in the structure of eugenol. This modified polymer is expected to be applied to pressure-sensitive adhesives and silicone rubber products for the biomedical field because of its reinforcing effect and antioxidant quality.

## Figures and Tables

**Figure 1 polymers-11-01389-f001:**
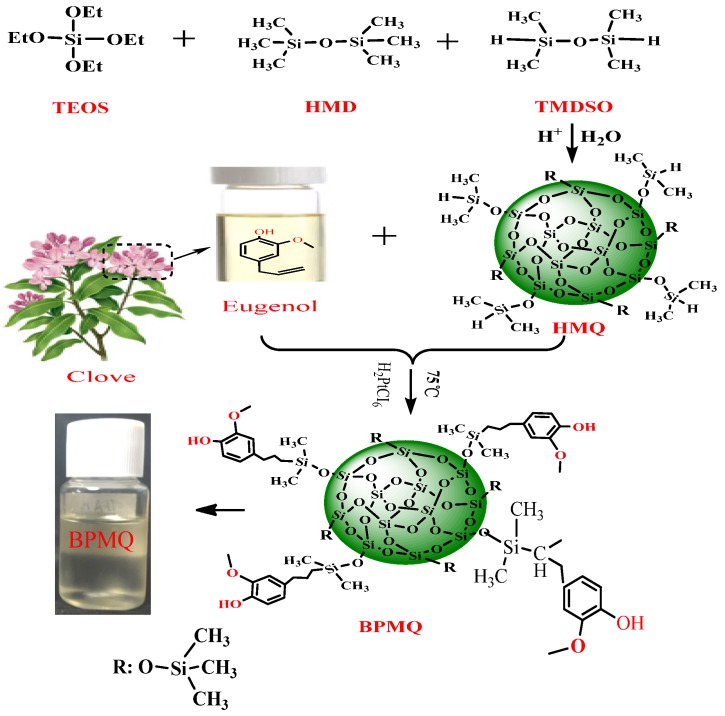
Scheme of synthesis of HMQ and BPMQ (TEOS: tetraethyl orthosilicate; HMD: hexamethyldisiloxane; TMDSO: 1,1,3,3-tetramethyldisiloxane; HMQ: hydrogen containing MQ silicone resin; BPMQ: bio-Phenol MQ Silicone Resin).

**Figure 2 polymers-11-01389-f002:**
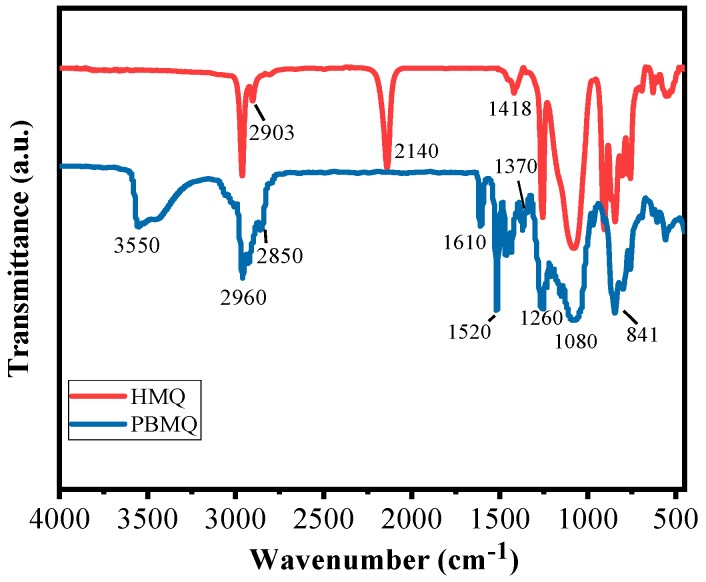
Infrared spectra of HMQ and BPMQ.

**Figure 3 polymers-11-01389-f003:**
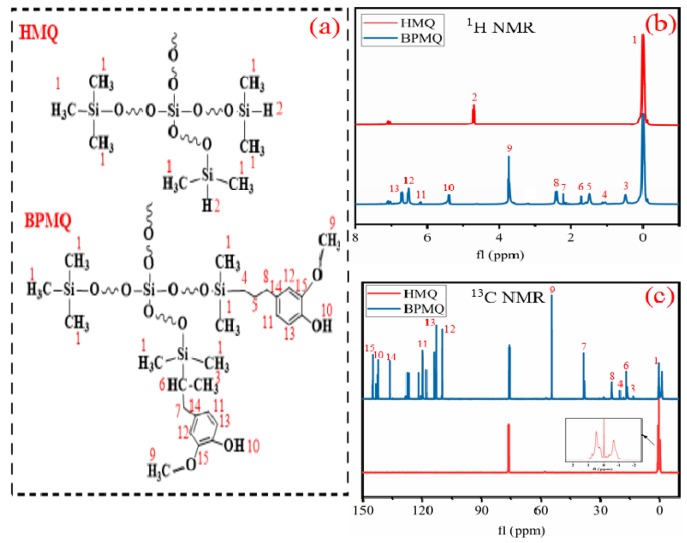
^1^H NMR and ^13^C NMR spectroscopic analysis. (**a**) Molecular formula of HMQ and BPMQ; (**b**) ^1^H NMR spectrum of HMQ and BPMQ; (**c**) ^13^C NMR spectrum of HMQ and BPMQ.

**Figure 4 polymers-11-01389-f004:**
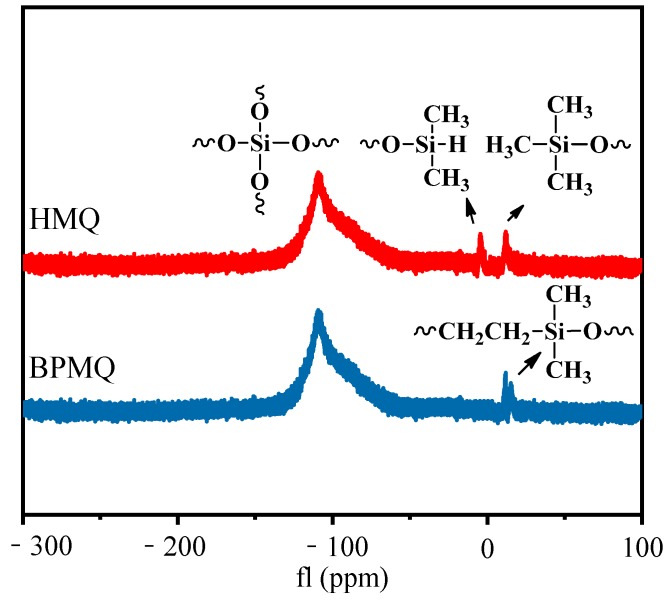
^29^Si NMR spectra of HMQ and BPMQ.

**Figure 5 polymers-11-01389-f005:**
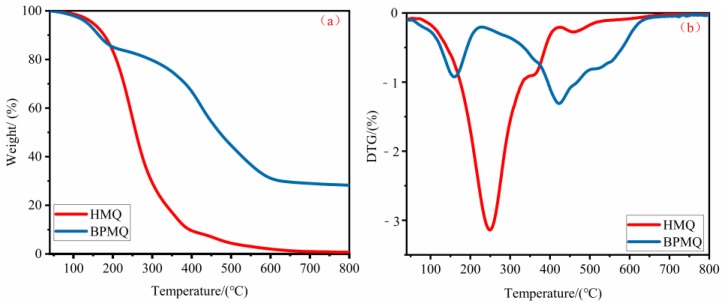
TG (**a**) and DTG (**b**) curves of HMQ and BPMQ, obtained in a nitrogen atmosphere.

**Figure 6 polymers-11-01389-f006:**
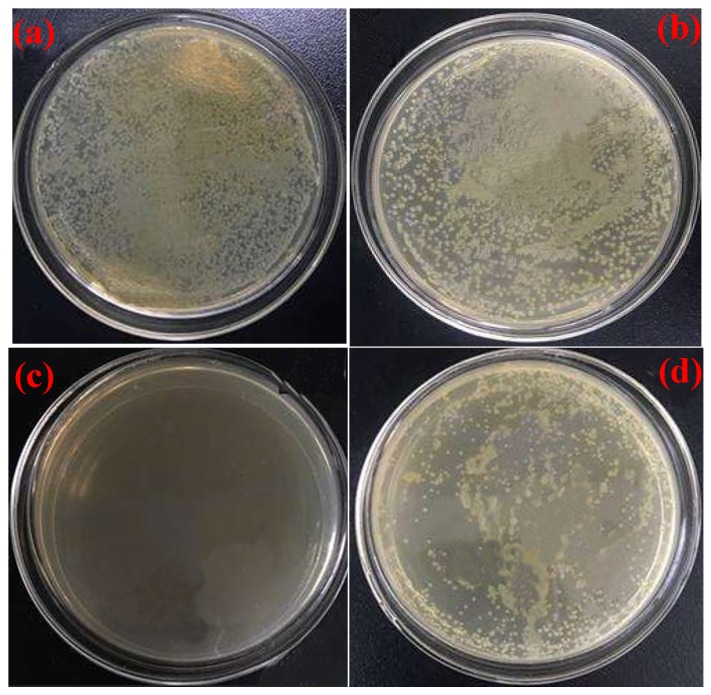
Optical images of agar plates showing the antimicrobial activity: the blank group (**a**), HMQ (**b**), eugenol (**c**), and BPMQ (**d**).

**Table 1 polymers-11-01389-t001:** The molecular weights and viscosity of HMQ and BPMQ.

Entry	*Mw* (dal.mol^−1^)	*Mn* (dal.mol^−1^)	*Mw/Mn*	*Dynamic Viscosity* (mpa.s)
HMQ	2000	1400	1.43	440
BPMQ	2600	1800	1.44	465

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
