# Peer review of "A Simple Preparation Route for Bio-Phenol MQ Silicone Resin via the Hydrosilylation Method and its Autonomic Antibacterial Property"

_polymers, 2019, doi:10.3390/polym11091389_

Round 1

Reviewer 1 Report

  The introduction of eugenol units into MQ silicone resin and its characterization were described. The authors also investigated the thermal and antibacterial properties. The manuscript contains interesting results, but the following points should be improved to publish it.

3.4. Antibacterial properties. The authors did not describe how much the HMQ, eugenol, and BPMQ were used. Especially, the information is very important to compare the activity of eugenol with that of BPMQ. Is the concentration of eugenol used equal to that of the eugenol units in BPMQ (Fig 6 c and d)? The results show that eugenol is more active than BPMQ. The amount of the eugenol units in BPMQ should be identical to the amount of eugenol. If so, the description should be added.

In Table 1. Molecular weights (Mn and Mw) of HMQ and BPMQ are estimated by GPC (PSt standards). The values are not absolute molecular weights. The significant figure should be two digits, ex., 2645 -> 2600.

In Abstract. What does MQ stand for? The authors need to add a brief explanation on it or use another name instead of MQ because many readers do not know what MQ is.

Line 33. What is “active hydrogen”? An “active hydrogen” is not an organic group.

Line 181. What is “curve b”? Does “curve b” mean blue line (PBMQ)?

Author Response

Response to Reviewer 1 Comments

Point 1:    Antibacterial properties. The authors did not describe how much the HMQ, eugenol, and BPMQ were used. Especially, the information is very important to compare the activity of eugenol with that of BPMQ. Is the concentration of eugenol used equal to that of the eugenol units in BPMQ (Fig 6 c and d)? The results show that eugenol is more active than BPMQ. The amount of the eugenol units in BPMQ should be identical to the amount of eugenol. If so, the description should be added.

Response 1: Thank you for your indication. The related description was added in section 2.4 and section 3.4,  correspond to “line 160 -163” and  “line 276-279”, respectively. In line 160 -163, as follows, “To more intuitive the antimicrobial effect caused by HMQ, PBMQ and eugenol, the same mass fraction of HMQ and BPMQ (equivalent to 20 % of the weight of agar) were added to the agar plates. Likewise, kept the amount of eugenol was identical to that amount of  eugenol units in BPMQ. ” and in line 276-279, as follows,“In order to more intuitive the antimicrobial effect caused by HMQ, PBMQ and eugenol, the same mass fraction of HMQ and BPMQ were added to the agar plates. simultaneously, kept the amount of eugenol was identical to that amount of eugenol units in BPMQ.”

Point 2: In Table 1. Molecular weights (Mn and Mw) of HMQ and BPMQ are estimated by GPC (PSt standards). The values are not absolute molecular weights. The significant figure should be two digits, ex., 2645 -> 2600.

Response 2: Thank you for your suggestion, The Molecular weights (Mn and Mw) of HMQ and BPMQ have been corrected to two significant digits. ex., 1951-> 2000ï¼›1352-> 1400ï¼›2645-> 2600ï¼›1794-> 1800 dal.mol-1.

Point 3: In Abstract. What does MQ stand for? The authors need to add a brief explanation on it or use another name instead of MQ because many readers do not know what MQ is.

 Response 3: Thank you for your reminder. The “MQ ”was stand for the combination of“M units”and “Q units”,in order to briefly explain the MQ silicone resin polymers, we have described them in Abstract, as follows,“MQ silicone resins are broad range of hydrolytic condensation products of monofunctional silane (M units) and tetrafunctional silane (Q units).”

Point 4: Line 33. What is “active hydrogen”? An “active hydrogen” is not an organic group.

Response 4: Thank you for your indication. The “active hydrogen” in Line 33 is “silicon hydrogen bond”,we have corrected it now,and have changed the sentence from “The core of MQ silicone resin is a cage type SiO2 (Q units) with high density, while the shell is M units with low density, and organic groups such as methyl, aryl, vinyl and active hydrogen  are existed in M units ” to “The core of MQ silicone resin is a cage type SiO2 (Q units) with high density, while the shell is M units with low density, and organic/inorganoic groups such as methyl, aryl, vinyl and silicon hydrogen bond are existed in M units”.

 Point 5: Line 181. What is “curve b”? Does “curve b” mean blue line (PBMQ)? 

Response 4: Thank you for your reminder. The “curve b” means the“ curve of PBMQ”.We have corrected it in the article,as follows,“The structure of HMQ and BPMQ were characterized by FT-IR. As is shown in Figure 2, in the curve of HMQ...”and “ In the curve of PBMQ, a broad peak appeared at 3550 cm-1 is corresponding to the stretching vibration of phenol hydroxyl.”

Reviewer 2 Report

The research work done by Jianfang Ge research group “Facile preparation Route for Bio-Phenol MQ silicone resin via hydrosilylation Method and its Autonomic Antibacterial Property” describes a good protocol for making Bio-Phenol MQ silicone resin. Authors documented preparation of Bio-Phenol MQ silicone resin using hydrosilylation conditions. The major advantage of the present documented report is the antibacterial property against E. Coli was increased without adding any other antimicrobial agents which makes the report attractive for readers. There are not many literature procedures, similar to present documented approach and these Bio-Phenol MQ silicone resins are important in Bio related field, also this approach has broad applicability for further application and development. Given the importance of practicality for this work, I recommend the publication of this manuscript in the Polymers.

Author Response

Response to Reviewer 2 Comments

Point 1:  The research work done by Jianfang Ge research group “Facile preparation Route for Bio-Phenol MQ silicone resin via hydrosilylation Method and its Autonomic Antibacterial Property” describes a good protocol for making Bio-Phenol MQ silicone resin. Authors documented preparation of Bio-Phenol MQ silicone resin using hydrosilylation conditions. The major advantage of the present documented report is the antibacterial property against E. Coli was increased without adding any other antimicrobial agents which makes the report attractive for readers. There are not many literature procedures, similar to present documented approach and these Bio-Phenol MQ silicone resins are important in Bio related field, also this approach has broad applicability for further application and development. Given the importance of practicality for this work, I recommend the publication of this manuscript in the Polymers.

Response 1: Thank you very much for your recognition of this research work. We have also corrected the deficiencies in the article according to the review  1 comments. Thanks again for your evaluation and recognition of this research work.

Round 2

Reviewer 1 Report

The revised manuscript is much better than the previuos one.

Publication is recommended.